# Solvent Effect on Small-Molecule Thin Film Formation Deposited Using the Doctor Blade Technique

Rodrigo Ramos-Hernández [1], Enrique Pérez-Gutiérrez [1], Francisco Domingo Calvo [2,*], Miriam Fatima Beristain [1], Margarita Cerón [1] and Maria Judith Percino [1,*]

1 Unidad de Polímeros y Electrónica Orgánica, Instituto de Ciencias, Benemérita Universidad Autónoma de Puebla, Val3-Ecocampus Valsequillo, Puebla 72960, Mexico
2 Decanatura de Ingenierías, Universidad Popular Autónoma de Puebla, Puebla 72410, Mexico
* Correspondence: franciscodomingo.calvo@upaep.mx (F.D.C.); judith.percino@correo.buap.mx (M.J.P.)

**Abstract:** Small molecule thin films are the core of some organic optoelectronic devices. Their deposition by solution processes is an advantage for device fabrication and can be achieved via spin coating for small areas and slot-die or doctor blade for larger areas. Solution deposition of small molecule thin films is usually processed only with medium polarity solvents. Herein, the use and influence of solvents with several polarities and physicochemical properties to form small-molecule homogeneous thin films via the doctor blade technique over an area of 25 cm$^2$ have been explored. Solvents with different polarity, heptane, chlorobenzene, N,N-dimethylformamide, acetonitrile, and methanol were used along with different deposition temperatures, from room temperature up to near the boiling temperature for each solvent. With heptane and chlorobenzene, smooth films with an average roughness of 3 nm and thickness of 100–120 nm were obtained. The film was homogeneous over the whole substrate for temperatures from room temperature to close to the boiling temperature of both solvents. On the other hand, with dimethylformamide, a film is observed when the deposition is conducted only at room temperature; when the deposition temperature increases, the formation of agglomerates of several sizes from 1 to 5 nm was observed. With acetonitrile, and methanol, no films were formed, and only nanoaggregates were created on the substrate due to the solvent high vapor pressure, and the agglomerate size depends on the deposition temperature. The measure of the contact angle of pure solvent and solutions indicated that wettability helps to film formation over the whole substrate. For heptane and chlorobenzene, a small angle was measured; meanwhile, the contact angle is large in acetonitrile leading to the formation of nanoaggregates. In the case of methanol solution, although it wets very well, no film is deposited because it has high volatility.

**Keywords:** small-molecule films; large area; solvent effect; doctor blade technique



## 1. Introduction

Thin films of organic semiconductors are the scaffold of several optoelectronic devices such as organic light-emitting diodes (OLEDs), organic solar cells (OPVs), or field-effect transistors (OFETs); in many cases, features of thin films determine the performance of the device. Different deposition techniques for thin films are feasible depending on the nature of the organic semiconductor. For polymers, the solution process is used because the chain entanglement assures film formation. In contrast, high vacuum evaporation is mainly used for small molecules, although recently solution deposition has also been used [1–3]. Today, many optoelectronic devices are based on small molecules firstly because of the versatility of the synthesis and secondly because of their optoelectronic properties, which allow tuning by modifying their molecular structure with functional groups [4–9]. The spin coating technique is the most common method for depositing organic semiconductor films from a solution, mainly used at the lab scale. The spin coating has the drawback of having a lot of solution waste and is incompatible with the roll-to-roll process for in-series production. Recently, the doctor blade (DB) method has been reported for small



molecule film deposition [10–17]; this method has several advantages compared with vacuum evaporation or spin coatings, for instance, a larger coverage area with low material waste and lower cost. In addition, DB allows scaling for a continuous process because it is compatible with the roll-to-roll process. Therefore, DB is used for device fabrication; for OLEDs, one or even all the organic layers in the structure can be deposited via the blade coating method [10–13]. You et al. reported the fabrication of all-solution-processed OLED; they conducted layer-by-layer film deposition with thicknesses of about 20–50 nm with no dissolution problem just by varying solvent. Therefore, the active layer, as well as injection and charge transport layers, were deposited via the doctor blade technique [10]. Moreover, Chang et al. used the blade coating technique for preparing phosphorescent OLEDs; they used a small molecule for the emissive and electron transport layers. They achieved a homogeneous film in an area of 6 cm$^2$, with roughness as low as 0.2 nm [11]. Despite the doctor blade technique being now used for the preparation of small molecule-based OLEDs, OPVs, and OFETs [10,12,14–17], little research has carried out to reveal the nature of the small molecule thin film formation via the blade technique. For polymers, chain entanglement is responsible for film formation, but this kind of interaction is not present in small molecules. It is well known for small molecule solutions; molecules tend to crystallize or agglomerate when the solvent evaporates; hence, a similar behavior instead a homogeneous film could be expected for the doctor blade technique.

Herein, the effect of solvent over the formation of small molecule films deposited by the doctor blade technique is reported; the films were deposited over an area of 25 cm$^2$. Solvents with different polarity and vapor pressure heptane, chlorobenzene, dimethylformamide, acrylonitrile, and methanol were chosen to prepare solutions for blade deposition. Several temperatures from room to above boiling temperature were used for each solvent. The morphology, roughness, and thickness of films were analyzed via atomic force microscopy. Solvents with low polarity heptane and chlorobenzene, but different vapor pressure, allow for obtaining smooth films with average roughness and thickness of about 3 nm and 100–120 nm, respectively. On the other hand, dimethylformamide leads to film formation only when the deposition is conducted at room temperature; with higher temperatures, agglomerates are obtained. Solvents with higher polarity and vapor pressure, acetonitrile, and methanol form nanoaggregates despite the deposition temperature. The contact angle for solvents and solutions was also measured and showed that solution-substrate interaction influences film formation.

## 2. Materials and Methods

### 2.1. Materials

The glass/ITO substrates with 10–15 Ω/sq were acquired from Delta Technologies, Loveland, CO, USA. Poly(3,4-ethylenedioxythiophene)-poly(styrenesulfonate) (PEDOT:PSS), PVP AI4083 was purchased from Heraeus-Clevios, Leverkusen, Germany. Solvents heptane, chlorobenzene, methanol, acetonitrile, and dimethylformamide were acquired from Sigma Aldrich (St. Louis, MO, USA). A family of small molecule acrylonitrile derivative:

(Z)-2-(4-bromophenyl)-3-(4-(diphenylamino)phenyl)acrylonitrile       (A),
(Z)-3-(4-(diphenylamino)phenyl)-2-phenylacrylonitrile       (B),
(Z)-2-(4-chlorophenyl)-3-(4-(diphenylamino)phenyl)acrylonitrile       (C),
(Z)-2-(4-(fluorophenyl)-3-(4-(diphenylamino)phenyl)acrylonitrile       (D)
(Figure S1) were synthetized according to previous reported method [18].

### 2.2. Film Deposition

Glass/ITO of 5 cm × 5 cm was used as substrate; it was cleaned with ethanol previously to the deposition. Each substrate was covered with a PEDOT:PSS layer, for which the polymer suspension was diluted with EtOH (1:3% *v/v*). Solutions of the compounds A-D were prepared, and the concentration was in accordance with the solubility of the compounds in each solvent: heptane (2.5 mg/mL), chlorobenzene (10 mg/mL), N,N-dimethylformamide (5 mg/mL), acetonitrile (4 mg/mL), or methanol (2 mg/mL). The deposition of the films was carried out via the doctor blade technique implemented with a

computer numerical control (CNC) machine. The blade was fixed in a shaft with movement in the *z*-axis. Meanwhile, a mobile platform x-y with movement in the *x*-axis was used as a substrate-holder. The platform displacement allows the spreading of the solution dispensed under the blade. For PEDOT:PSS and small-molecule films, the substrate–blade gap was set at 200 μm by using a mechanical separator. The coating speed was fixed at 500 mm/min for all the experiments. For PEDOT:PSS films, the substrate was heated to 50 °C and then placed at the mobile platform of the blade coater, 100 μL of PEDOT:PSS solution was dispensed at the applicator–substrate gap, and the platform was moved under the blade spreading the solution. Therefore, a homogeneous film (40 nm thickness) was obtained. For small molecule films, 80 μL of each solution is dispersed at the applicator–substrate gap, then bladed onto the substrate. The temperature for deposition was varied from room temperature (RT) up to near the boiling point of each solvent (Table 1). After blading process, substrates were placed on a hot plate for annealing at 50 °C.

**Table 1.** Conditions for the solutions of compounds A–D in each solvent and temperature used for the film deposition via the doctor blade technique.

| Solvent | Relative Polarity | Vapor Pressure (hPa) | Boiling Point (°C) | Concentration (mg/mL) | Deposition Temperature (°C) | Deposit Features |
|---|---|---|---|---|---|---|
| Heptane | 0.012 | 48 | 98 | 2.5 | RT | Dense and homogeneous film |
| | | | | | 50 | |
| | | | | | 75 | |
| | | | | | 90 | |
| Chlorobenzene | 0.188 | 12 | 132 | 10 | RT | Dense and homogeneous film |
| | | | | | 50 | |
| | | | | | 100 | Agglomerates |
| | | | | | 150 | |
| DMF | 0.386 | 3.5 | 153 | 5 | RT | Dense and homogeneous film |
| | | | | | 50 | Non total coverage |
| | | | | | 100 | Small agglomerates |
| | | | | | 150 | Dense and homogeneous film |
| Acetonitrile | 0.46 | 97 | 81.6 | 4 | RT | Small agglomerates |
| | | | | | 50 | |
| | | | | | 85 | |
| | | | | | 110 | |
| Methanol | 0.762 | 128 | 64.6 | 2 | RT | Small agglomerates |
| | | | | | 50 | |
| | | | | | 65 | |
| | | | | | 90 | |

### 2.3. Characterization

The morphology and thickness were analyzed via atomic force microscopy (AFM) with EasyScan2 equipment from Nanosurf (Liestal, Switzerland, software version 3.10.0), operating in contact mode under ambient conditions. For contact angle measurement, a setup with a camera EOS REBEL T6, Canon, (New York, NY, USA). was used. The substrate (glass/ITO/PEDOT:PSS) was placed on a hotplate to set the temperature of the experiment. Ten microliters were dispensed onto the substrate to form a drop of each solution, and a picture was taken.

### 3. Results and Discussion

Films were deposited for each compound with the solvents: heptane, chlorobenzene, DMF, acetonitrile, and MeOH. Solvents were select according to polarity and vapor pressure

(Table 1). Using each solvent leads to obtaining films with different features and, in some cases, forming agglomerates instead of a homogeneous film. AFM was used to analyze the morphology of films or aggregates. A picture under UV light is also shown (inset in AFM images) for those deposits forming homogeneous films. Overall, the behavior in the film formation for compounds A–D was similar; therefore, only the results for the compound (Z)-2-(4-bromophenyl)-3-(4-(diphenylamine)phenyl) acrylonitrile (B) are presented here; meanwhile, the result for compounds A, C, and D are given in the Supplementary Materials. Compound B has a molar mass of 451.37 gr/mol, which is below other molecules that have been deposited via the blade technique [11,13,14,19,20]. Interestingly, solvents with lower polarity, heptane, and chlorobenzene formed homogeneous and dense films over the whole substrate at room temperature and for most of the temperatures used for deposition (Table 1), as shown via AFM analysis and pictures of the films under UV light (Figure 1). The average roughness and thickness for films deposited from heptane solution are 3.8 nm and 107 nm, respectively. Meanwhile, in chlorobenzene, the average roughness and thickness are 2.8 nm and 122.6 nm, respectively; these values are similar to the report for the same deposition technique with other small molecules [10,12,14–17]. Heptane is not commonly used as a solvent for organic film deposition, which could be due to the poor solubility of many organic compounds in this solvent. In the present study, heptane gave the best results, i.e., the most homogeneous and smooth films (Figure 1). For chlorobenzene, homogeneous films are obtained when deposited at RT and 50 °C. Chlorobenzene is the most common solvent for organic compounds, for either spin coating or doctor blade technique; in many reports, both the solution and substrate need to be heated at 30–70 °C before deposition [21–25]. The temperature is commonly used to induce solvent evaporation. It is well known that, for several deposition techniques, the solvent evaporation rate is a critical parameter that affects the thickness and homogeneity of films. In the present work, when the temperature increases to 100 °C, aggregates are formed instead of a homogeneous film owing to the accelerated solvent evaporation (Figure 1), which was verified when the temperature was increased to 150 °C, and bigger aggregates were observed. Similar behavior was observed for compounds A, C, and D as shown in Figures S1, S3, and S5, respectively.

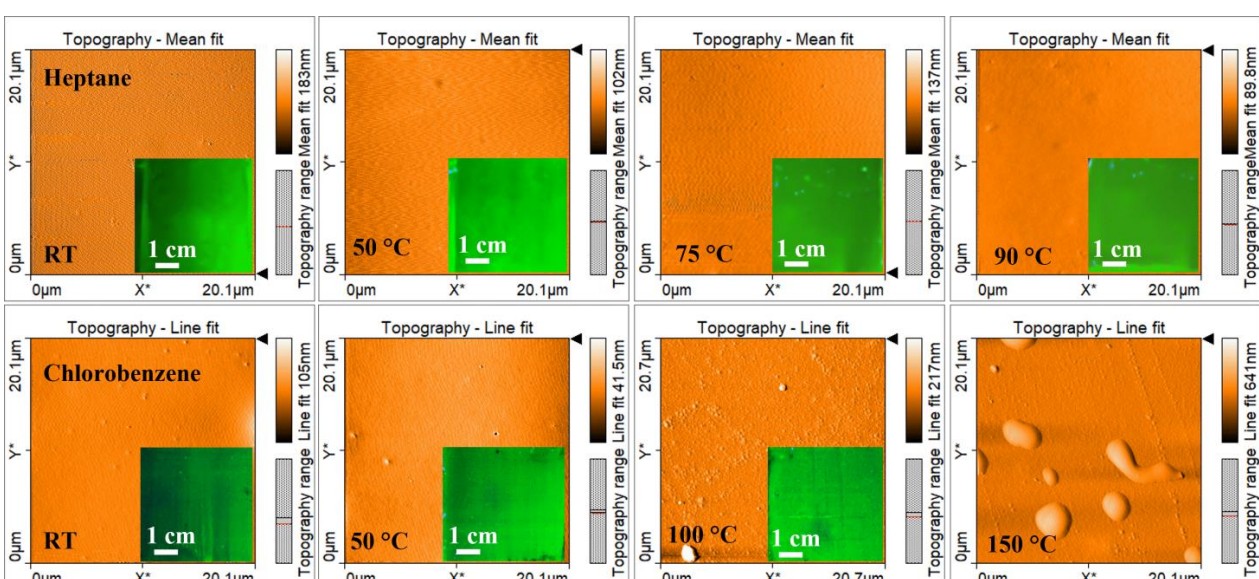

**Figure 1.** Representative AFM images of compound B films deposited by the blade coating technique with heptane and chlorobenzene as solvent. The inset is pictures of the films under UV light.

On the other hand, when the solvent polarity changes by using DMF, films with coverage over the whole substrate are obtained only at RT, while agglomerates of several sizes were formed when the temperature rose. For RT, the solution is bladed over the

substrate surface, and owing to the high boiling point of DMF, no solvent evaporation occurs until the substrate is transferred to the hot plate. Therefore, despite having a complete spreading of solution over the whole substrate, a non-homogeneous film with thicker and thinner zones is obtained (inset in RT Figure, Figure 2). When the substrate and DMF solution are heated before deposition, aggregates are formed onto the substrate instead of a homogeneous film. Solution deposited at 50 °C showed larger aggregates in the range of micrometers; at this temperature, solvent evaporation has begun but not fast enough, allowing larger drops onto the surface. Increasing the temperature reduces the aggregate size because it also increases the solvent evaporation rate; at 100 °C and 150 °C, films composed of nanoaggregates were observed (Figure 2).

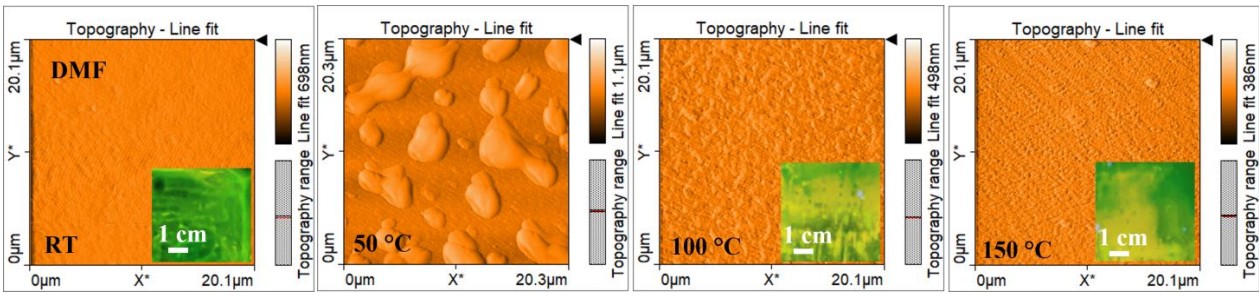

**Figure 2.** Representative AFM image for films deposited with DMF as solvent. The inset is pictures of the films under UV light.

In the case of acetonitrile and MeOH, aggregates were formed on the surface independently of the deposition temperature (Figures 3 and S2, S4, and S6). As observed with DMF, the acetonitrile and MeOH aggregates are smaller as the temperature increases. In this case, the observed morphology could be due to faster solvent evaporation because of the high vapor pressure, which avoids film formation.

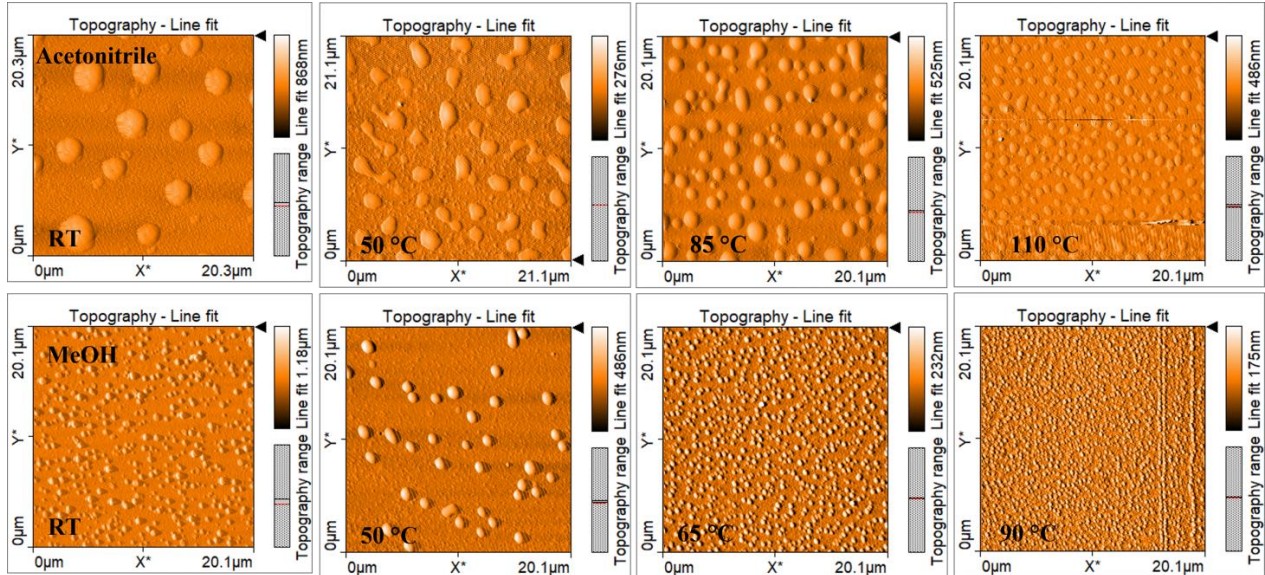

**Figure 3.** Representative AFM image for aggregates formed from acetonitrile and MeOH solutions.

The solution–substrate interaction also influences the film's formation and morphology. The wettability of each solvent on the surface could lead to total coverage or agglomerate formation. Therefore, the contact angle for pure solvents and solutions with the dye was measured, and the values were almost the same for both cases (Figure 4). Heptane showed good wettability, and no angle was observed after drop deposition; meanwhile, chlorobenzene and DMF showed small contact angles of about 20° and 16°, respectively. The good

wettability allows the solution to spread over the whole substrate surface before solvent evaporation, and a homogeneous film is obtained. Meanwhile, for acetonitrile, a larger angle was measured at about 43°. The observed morphology, i.e., aggregate formation by using acetonitrile, could arise from the non-wettability of this solvent, avoiding the solution spread and total coverage onto the substrate. On the other hand, MeOH showed good wettability; however, it is the solvent with higher vapor pressure. Therefore, despite a total spreading of MeOH solution, the fast solvent evaporation leads to aggregates instead a homogeneous film.

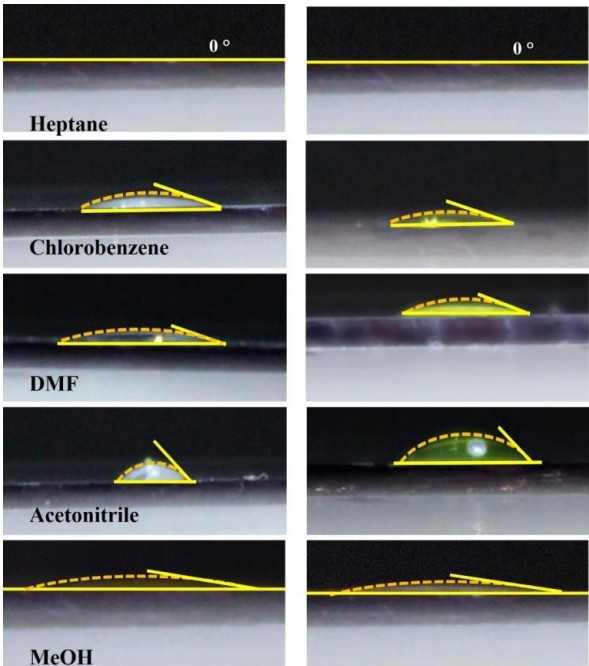

**Figure 4.** Contact angle for pure solvents (**left**) and solution used for film deposition (**right**).

It is known that solvent evaporation has a strong influence on film formation, even for polymers. For instance, when the solvent is rapidly evaporated, Marangoni instabilities can be created in polymer films deposited via the spin coating technique. In the Marangoni process, gradients in surface tension (owing to solvent evaporation) drive the formation of convection cells, which ultimately lead to variations in the roughness and thickness of the deposited film [26]. Once the surface tension gradient has been established, the solution begins to flow from the low surface tension region toward the high surface tension area. Thus, the region of high surface tension will pull material from the area with lower surface tension, leading to the formation of convection cells [27,28]. In contrast, no Marangoni flow is seen for slowly evaporating solvents, resulting in smooth surfaces [26,27]. The convection cells owing to the Marangoni effect are observed as "hills" and "valleys" in polymer films; the "valleys" are, in fact, a film with a thinner thickness. The entanglement of polymer chains allows this thinner film. For small-molecule thin films, the formation of convection cells could be the reason for aggregate formation instead a homogeneous film with solvents with higher vapor pressure, such as acetonitrile and methanol. In addition, it has been reported that when the spreading time for solutions in volatile solvents is larger than the solvent evaporation time, the solutions do not spread completely and thus yield large contact angles [26]. The results here reported for films of low molecular weight compounds deposited via the doctor blade technique indicated that, for small molecules, the most important parameters for film formation are those related to solvent instead the dye. This behavior has also been observed for other small molecule solutions' physiochemical parameters. For instance, Feng et al. reported a viscosity study for small molecular solutions; they found that the viscosity of solvent mainly determines the solution

viscosity value and no significant changes with increasing solution concentration; this behavior is quite different from that of polymer [29].

## 4. Conclusions

The doctor blade technique allows the formation of small molecule thin films over an area of 25 cm$^2$. For deposition, solvents with different polarity and vapor pressure were used. It was found that smooth films can be deposited even at room temperature for solvents with low vapor pressure heptane and chlorobenzene. However, by increasing the polarity and vapor pressure, aggregates were formed instead of a dense film; this behavior was for acrylonitrile and methanol. The temperature also influences film formation. When the deposition is conducted at temperatures near the solvent boiling point, homogeneous films can be changed to agglomerates, and the size of agglomerates decreases as the temperature increases. The results showed that contrary to polymers where film formation is attributed to chain entanglement, in small-molecule thin films, the solvent and not molecule features could define the film's formation.

**Supplementary Materials:** The following supporting information can be downloaded at: https://www.mdpi.com/article/10.3390/coatings13020425/s1; Figure S1: Chemical structures of compounds (a) I, (b) II, (c) III and (d) IV; Figure S2: Representative AFM images of films obtained whit compound II for heptane, chlorobenzene and DMF at different temperatures; Figure S3: Representative AFM images of deposits obtained whit compound II for acetonitrile and methanol at different temperatures; Figure S4: Representative AFM images of films obtained whit compound III for heptane, chlorobenzene and DMF at different temperatures; Figure S5: Representative AFM images of deposits obtained whit compound III for acetonitrile and methanol at different temperatures; Figure S6: Representative AFM images of films obtained whit compound IV for heptane, chlorobenzene and DMF at different temperatures; Figure S7: Representative AFM images of deposits obtained whit compound IV for acetonitrile and methanol at different temperatures.

**Author Contributions:** Conceptualization, R.R.-H., E.P.-G., F.D.C. and M.J.P.; Funding acquisition, E.P.-G. and M.J.P.; Writing—original draft, R.R.-H., F.D.C. and M.C.; Writing—review & editing, E.P.-G., M.F.B., M.C. and M.J.P. All authors have read and agreed to the published version of the manuscript.

**Funding:** This research was funded by VIEP-BUAP, project 00110-VIEP-2022 and CONACYT project CF-2019-51472.

**Institutional Review Board Statement:** Not applicable.

**Informed Consent Statement:** Not applicable.

**Data Availability Statement:** Not applicable.

**Acknowledgments:** To CATEDRAS-CONACYT.

**Conflicts of Interest:** The authors declare no conflict of interest.

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
