# Peer review of "Solvent Effect on Small-Molecule Thin Film Formation Deposited Using the Doctor Blade Technique"

_coatings, doi:10.3390/coatings13020425_

Round 1

Reviewer 1 Report

In this manuscript, the use and effect of solvents with several physicochemical properties to fabricate large-area thin films using doctor blade method were investigated. It was observed that the deposition temperature, the polarity and vapor pressure of chemicals effect the film morphology. The results are interesting and valuable. As such, the paper may be of interest to the readers of Coatings. However, I think the article has several deficiencies and it requires some corrections before being considered for publication.

1)    Please, create a table containing the vapor pressure of each chemical used in film production.

2)    Please add the volume of water used in contact angle measurements in experimental part.

3)    Try to explain the reason for the effect of vapor pressure on surface morphology. Mention other studies in the literature showing the effect of vapor pressure on surface morphology, even if with other film fabrication methods.

Author Response

Reviewer 1

Comments and Suggestions for Authors

In this manuscript, the use and effect of solvents with several physicochemical properties to fabricate large-area thin films using doctor blade method were investigated. It was observed that the deposition temperature, the polarity and vapor pressure of chemicals effect the film morphology. The results are interesting and valuable. As such, the paper may be of interest to the readers of Coatings. However, I think the article has several deficiencies and it requires some corrections before being considered for publication.

1)    Please, create a table containing the vapor pressure of each chemical used in film production.

Answer:

According to the reviewer's comment, on page 4, Table 1 was modified by adding the relative polarity, vapor pressure, and boiling point for the used solvents. The added values are:

 Physicochemical parameters of the used solvents.

Solvent

Relative polarity

Vapor pressure

(hPa)

Boiling point

(°C)

Heptane

0.012

48

98

Chlorobenzene

0.188

12

132

DMF

0.386

3.5

153

Acetonitrile

0.46

97

81.6

Methanol

0.762

128

64.6

2)    Please add the volume of water used in contact angle measurements in experimental part.

Answer:

The drop volume for the contact angle measurement was about ten μl, dispensed with a micropipette. To add this information in the manuscript, the sentence in section 2.3, page 3:

A drop of solution was deposited onto the substrate, and a picture was taken.”

Was rewritten as follow:

Ten microliters were dispensed onto the substrate to form a drop of each solution, and a picture was taken.”

3)    Try to explain the reason for the effect of vapor pressure on surface morphology. Mention other studies in the literature showing the effect of vapor pressure on surface morphology, even if with other film fabrication methods.

Answer:

The effect of the solvent evaporation rate (which is proportional to the solvent vapor pressure) on the morphology and roughness of polymer film deposited by the spin coating technique has been extensively studied. Two models have been proposed to explain the formation of inhomogeneities on the surface of polymer films. The first model suggests that rapid solvent evaporation yields a dry polymer skin on the free surface and a fluid solution underneath. The solvent continues to evaporate via diffusion through the dry skin, and when all the solvent has evaporated, the polymer skin buckles to form a corrugated surface. [1] The second model suggests that the inhomogeneities are due to the Marangoni effect. In the case of a Marangoni process, gradients in surface tension (owing to solvent evaporation) drive the formation of convection cells, which ultimately lead to variations in the film thickness. [2] Surface tension gradients may arise due to gradients in temperature or concentration. In either case, once the surface tension gradient has been established, the solution begins to flow from the low surface tension region toward the high surface tension area. Thus, the region of high surface tension will pull material from the area with lower surface tension, leading to the formation of convection cells. [3]

[1] L. Pauchard, C. Allain, Buckling instability induced by polymer solution drying, Europhys. Lett. 62 (2003) 897–903. http://doi.org/10.1209/epl/i2003-00457-7.

[2] P.D. Fowler, C. Ruscher, J.D. McGraw, J.A. Forrest, K. Dalnoki-Veress, Controlling Marangoni-induced instabilities in spin-cast polymer films: How to prepare uniform films, Eur. Phys. J. E. 39 (2016) 90. https://doi.org/10.1140/epje/i2016-16090-9.

[3] D.P. Birnie III, Rational solvent selection strategies to combat striation

formation during spin coating of thin films, J. Mater. Res. 16 (2001) 1145–1154. https://doi.org/10.1557/JMR.2001.0158.

In the present work, we have described the effect of vapor pressure on surface morphology in the last paragraph of the results section as follows:

It is known that solvent evaporation has a strong influence on film formation, even for polymers. For instance, when the solvent is rapidly evaporated, Marangoni instabilities can be created in polymer films deposited by the spin coating technique. In contrast, no Marangoni flow is seen for slowly evaporating solvents, resulting in smooth surfaces. Therefore, roughness can be controlled by varying the solvent evaporation rate in polymer films. [26,27] In addition, it has been reported that when the spreading time for solutions in volatile solvents is larger than the solvent evaporation time, the solutions do not spread completely and thus yield large contact angles. [26,27]

According to suggestion the paragraph page 6, was rewritten:

“It is known that solvent evaporation has a strong influence on film formation, even for polymers. For instance, when the solvent is rapidly evaporated, Marangoni instabilities can be created in polymer films deposited by the spin coating technique. In the Marangoni process, gradients in surface tension (owing to solvent evaporation) drive the formation of convection cells, which ultimately lead to variations in the roughness and thickness of the deposited film. [26] Once the surface tension gradient has been established, the solution begins to flow from the low surface tension region toward the high surface tension area. Thus, the region of high surface tension will pull material from the area with lower surface tension, leading to the formation of convection cells. [26–28] In contrast, no Marangoni flow is seen for slowly evaporating solvents, resulting in smooth surfaces. [26,27] The convection cells owing to the Marangoni effect, are observed as “hills” and “valleys” in polymer films; the “valleys” are, in fact, a film with a thinner thickness. The entanglement of polymer chains allows this thinner film. But for small-molecule thin films, the formation of convection cells could be the reason for aggregates formation instead a homogeneous film with solvents with higher vapor pressure, such as acetonitrile and methanol. In addition, it has been reported that when the spreading time for solutions in volatile solvents is larger than the solvent evaporation time, the solutions do not spread completely and thus yield large contact angles. [26]

It should be worth noting that the descriptions mentioned above of solvent effect on film morphology are for polymers, and to our knowledge, no similar theories are for low molecular weight compounds. For small-molecule solutions usually the solvent is not considered. Therefore, along with a more detailed description of solvent effect on morphology of small-molecule thin film, in the conclusion section, page 9, line 52, the next sentence was added:

“The results showed that contrary to polymers where film formation is attributed to chain entanglement, in small-molecule thin films, the solvent and not of molecule features could define the film's formation.”

Reviewer 2 Report

Review Report

The manuscript “Solvent Effect on Small-Molecule Thin Films Formation Deposited by the Doctor-Blade Technique” reports the employment of the doctor-blade technique to deposit small molecule thin films, and the effect of the solvents on the film formation. Results presented in this manuscript have shown that for small molecule thin films prepared by the doctor-blade technique, the most important parameters are related to the solvents, such as their polarity, boiling point, and wettability. These results are organized in an easy-to-understand fashion, which will provide a guide for preparing small-molecule thin coatings. The manuscript will be of interest to the readership of Coatings, thus, I would personally support the publication of this manuscript in the journal of Coatings. Below are some minor revisions for polishing the current manuscript.

1. A table listing the differences between these five selected solvents is suggested to demonstrate why these solvents are selected, and how the properties of each solvent play a role in forming small-molecule thin films.

2. What is the “CNC machine”, line 100?

3. The description of the film features of Chlorobenzene at 100 °C (aggregates are formed, lines 142-143) does not match with the records in Table 1 and Figure 1. Please check it.

4. The scale bar of the pictures of the films under UV light (the insertS of the AFM images).

5. What are the color bar of the AFM images? The labels in these AFM images are a little bit vague.

6. Some pictures under UV light are missing, for example, Chlorobenzene, 150 °C in Figure 1, DMF 50 °C in Figure 2, and Figure 3.

7. The labels in Figure 3 are suggested to align with those in Figures 1 and 2.

8. Artifacts seem to dominate in some AFM images, like Heptane, 75 °C in Figure 1.

9. An more comprehensive discussion is suggested (lines 191-205), to analyze how the solvent’s evaporation influences the thin film formation, in terms of the selection of solvents and the deposition temperature.

Author Response

Reviewer 2

Comments and Suggestions for Authors

Review Report

The manuscript “Solvent Effect on Small-Molecule Thin Films Formation Deposited by the Doctor-Blade Technique” reports the employment of the doctor-blade technique to deposit small molecule thin films, and the effect of the solvents on the film formation. Results presented in this manuscript have shown that for small molecule thin films prepared by the doctor-blade technique, the most important parameters are related to the solvents, such as their polarity, boiling point, and wettability. These results are organized in an easy-to-understand fashion, which will provide a guide for preparing small-molecule thin coatings. The manuscript will be of interest to the readership of Coatings, thus, I would personally support the publication of this manuscript in the journal of Coatings. Below are some minor revisions for polishing the current manuscript.

  1. A table listing the differences between these five selected solvents is suggested to demonstrate why these solvents are selected, and how the properties of each solvent play a role in forming small-molecule thin films.

Answer:

According to the reviewer's comment, on page 4 of revised manuscript, Table 1 was modified by adding the relative polarity, vapor pressure, and boiling point for the used solvents. The added values are:

 Physicochemical parameters of the used solvents.

Solvent

Relative polarity

Vapor pressure

(hPa)

Boiling point

(°C)

Heptane

0.012

48

98

Chlorobenzene

0.188

12

132

DMF

0.386

3.5

153

Acetonitrile

0.46

97

81.6

Methanol

0.762

128

64.6

The solvents in the present work were selected to have those with low, medium, and high polarity and vapor pressure. This was mentioned on page 3, lines 126-127 of the revised manuscript as follows:

“Solvents were selected according to polarity and vapor pressure (Table 1).”

  1. What is the “CNC machine”, line 100?

Answer:

We agree and thank the reviewer; the term CNC was not defined in the manuscript. CNC is the acronym for computer numerical control. A “CNC machine” is a motorized station for handling tools and often a motorized maneuverable platform; a computer controls both stations according to specific input instructions. In the present work, a CNC machine was used to implement the doctor blade technique. The blade was fixed in a shaft with movement in the z-axis. Meanwhile, the mobile platform x-y with movement in the x-axis was used as a substrate-holder. This displacement allows the spreading of the solution dispensed under the blade.

In the manuscript, the sentence in page 3, lines 90-104 of revised manuscript:

“The films deposition was conducted with a CNC machine designed and implemented for the study”

Was rewritten as follow:

“The deposition of the films was conducted by the doctor blade technique implemented with a computer numerical control (CNC) machine. The blade was fixed in a shaft with movement in the z-axis. Meanwhile, a mobile platform x-y with movement in the x-axis was used as a substrate-holder. The platform displacement allows the spreading of the solution dispensed under the blade.”

  1. The description of the film features of Chlorobenzene at 100 °C (aggregates are formed, lines 142-143) does not match with the records in Table 1 and Figure 1. Please check it.

Answer:

We agree with the reviewer, in table 1 the film deposited at 100 °C was described as a “Dense and homogeneous film,” but this is a mistake because, on page 4, such film is described as: “In the present work, when the temperature increases at 100 °C, aggregates are formed instead of a homogeneous film”; which agrees with what is observed in Figure 1.

Therefore, in Table 1, the description for films deposited at 100°C was changed from:

“Dense and homogeneous film”

To

“Agglomerates”

Therefore, this description in the table now agrees with what is mentioned on page 4, lines 148-149, and observed in Figure 1.

  1. The scale bar of the pictures of the films under UV light (the insertS of the AFM images).

Answer:

For Figures 1 and 2, a scale bar was added to the film under UV light (the inset of the AFM images). The same was done for images in the Supplementary Information.

  1. What are the color bar of the AFM images? The labels in these AFM images are a little bit vague.

Answer:

The color bar on the AFM images indicates the height difference from the lower point (black color) to the higher end (white color) on the analyzed morphology. The color bar is usually indicated for AFM analysis.

The line fit or mean fit labels refers to an adjustment from the software to aid a better visualization of the morphology image.

  1. Some pictures under UV light are missing, for example, Chlorobenzene, 150 °C in Figure 1, DMF 50 °C in Figure 2, and Figure 3.

Answer:

The reviewer's observation is pertinent. In the manuscript, pictures of films under UV light are presented for deposits forming a film; this is the case for deposition with heptane, chlorobenzene, and DMF (at specific temperatures). When aggregates are formed, no pictures are presented because a film is not appreciated under UV light. The next is a representative example of such films.

However, we agree that this point should be clarified in the manuscript. Therefore, the sentences on page 3, lines 124-126:

“Films were deposited for each compound with the solvents: heptane, chlorobenzene, DMF, acetonitrile, and MeOH. Using each solvent leads to obtaining films with different features and, in some cases, forming agglomerates instead of a homogeneous film.”

Was rewritten as follow:

“Films were deposited for each compound with the solvents: heptane, chlorobenzene, DMF, acetonitrile, and MeOH. Solvents were select according to polarity and vapor pressure (Table 1). Using each solvent leads to obtaining films with different features and, in some cases, forming agglomerates instead of a homogeneous film. AFM was used to analyze the morphology of films or aggregates. A picture under UV light is also shown (inset in AFM images) for those deposits forming homogeneous films.”

  1. The labels in Figure 3 are suggested to align with those in Figures 1 and 2.

Answer:

Labels in Figure 3 were aligned with those in Figure 1 and 2.

  1. Artifacts seem to dominate in some AFM images, like Heptane, 75 °C in Figure 1.

Answer:

We agree with the reviewer. For smoother films with roughness values of about 1-10 nm, the noise-to-signal ratio of AFM measurement becomes higher. Therefore, smooth surfaces such as those of films deposited with heptane look to be dominated by artifacts. However, no special treatment was used on images to show the original morphology features.

  1. An more comprehensive discussion is suggested (lines 191-205), to analyze how the solvent’s evaporation influences the thin film formation, in terms of the selection of solvents and the deposition temperature.

Answer:

The last paragraph of results section on page 8, lines 206-223 of revised manuscript was rewritten to improve the discussion about the solvent evaporation effect on films formation. Therefore, the original paragraph:

It is known that solvent evaporation has a strong influence on film formation, even for polymers. For instance, when the solvent is rapidly evaporated, Marangoni instabilities can be created in polymer films deposited by the spin coating technique. In contrast, no Marangoni flow is seen for slowly evaporating solvents, resulting in smooth surfaces. Therefore, roughness can be controlled by varying the solvent evaporation rate in polymer films. [26,27] In addition, it has been reported that when the spreading time for solutions in volatile solvents is larger than the solvent evaporation time, the solutions do not spread completely and thus yield large contact angles. [26,27]

Was rewritten as follow:

“It is known that solvent evaporation has a strong influence on film formation, even for polymers. For instance, when the solvent is rapidly evaporated, Marangoni instabilities can be created in polymer films deposited by the spin coating technique. In the Marangoni process, gradients in surface tension (owing to solvent evaporation) drive the formation of convection cells, which ultimately lead to variations in the roughness and thickness of the deposited film. [26] Once the surface tension gradient has been established, the solution begins to flow from the low surface tension region toward the high surface tension area. Thus, the region of high surface tension will pull material from the area with lower surface tension, leading to the formation of convection cells. [26–28] In contrast, no Marangoni flow is seen for slowly evaporating solvents, resulting in smooth surfaces. [26,27] The convection cells owing to the Marangoni effect, are observed as “hills” and “valleys” in polymer films; the “valleys” are, in fact, a film with a thinner thickness. The entanglement of polymer chains allows this thinner film. But for small-molecule thin films, the formation of convection cells could be the reason for aggregates formation instead a homogeneous film with solvents with higher vapor pressure, such as acetonitrile and methanol. In addition, it has been reported that when the spreading time for solutions in volatile solvents is larger than the solvent evaporation time, the solutions do not spread completely and thus yield large contact angles. [26]

Reviewer 3 Report

1. Introduction. it is declared that homogeneous covered area is lower than 1 cm2 for spin coating in Line 46 Page 2. Where do the authors get such a precise value? From refs or your experiment?

2. Line 48 Page 2, Recently doctor blade (DB) method has been reported for small molecule film deposition. Refs should be provided.

3. Too many words of BUT in the manuscript.

4. Line 70, Page 2. Parameters such as vapor pressure and boiling temperature for the solvents should be provided.

5. Line 94, Page 2. they were should be it was.

6. Line 175, Page 5. What will happen if the substrate is treated with plasma to enhance the wetting with the droplet of acetonitrile?

7. Line 210. Do you use the solvent of chloroform?

Author Response

Reviewer 3

Comments and Suggestions for Authors

  1. Introduction. it is declared that “homogeneous covered area is lower than 1 cm2” for spin coating in Line 46 Page 2. Where do the authors get such a precise value? From refs or your experiment?

Answer:

We agree with the reviewer; the sentence “homogeneous coverage area is lower than 1 cm2 for spin coating technique” is not entirely precise. It is known that the thickness of films deposited by spin-coating varies from the center to the whole of the substrate; however, homogeneity can be achieved within 1-4 cm2. In this respect, some studies have been conducted.  For instance, the thickness homogeneity was studied by J.A. Britten and I.M. Thomas, who stated: “However, most of the film thickness variation occurs in the first 1–2 cm from the substrate center, leaving the remainder almost uniform.” [Journal of Applied Physics 71, 972 (1992); https://doi.org/10.1063/1.351323]

Also, Q. Liu et al., conducted a study of uniformity of spin coating film thickness on large area rectangular substrates. They showed a homogeneous thickness within the first 1-2 cm from the center of substrate, and a distribution of thicknesses over a 24x48 cm substrate.

[Coatings 2022, 12(9), 1253; https://doi.org/10.3390/coatings12091253]

On the other hand, the intention for mentioning the thickness variation is to highlight some drawbacks of the spin coating technique, primarily to be implemented in large areas or scale production. Therefore, the sentence on pages 1-2, lines 44-47:

“The most common method for the deposition of organic semiconductors films from a solution is the spin coating technique, which is mainly used at the lab scale because the homogeneous covered area is lower than 1 cm2, and there is also a lot of waste of solution”

Was rewritten as follow:

“The spin coating technique is the most common method for depositing organic semiconductors films from a solution, mainly used at the lab scale. The spin coating has the drawbacks of a lot of solution waste and is incompatible with the roll-to-roll process for in-series production.”

  1. Line 48 Page 2, “Recently doctor blade (DB) method has been reported for small molecule film deposition”. Refs should be provided.

Answer:

References 10-17 report the deposition of small molecule thin films by the doctor blade technique. Some of these references are described in the first paragraph of page 2, lines 52-61 of the revised manuscript. Therefore, these references were placed at the end of the sentence “Recently doctor blade (DB) method has been reported for small molecule film deposition [10-17]” on page 2, line 48.

  1. Too many words of “BUT” in the manuscript.

Answer:

The manuscript was revised to amend this syntax error.

  1. Line 70, Page 2. Parameters such as vapor pressure and boiling temperature for the solvents should be provided.

Answer:

According to the reviewer's comment, on page 4, Table 1 was modified by adding the relative polarity, vapor pressure, and boiling point for the used solvents. The added values are:

 Physicochemical parameters of the used solvents.

Solvent

Relative polarity

Vapor pressure

(hPa)

Boiling point

(°C)

Heptane

0.012

48

98

Chlorobenzene

0.188

12

132

DMF

0.386

3.5

153

Acetonitrile

0.46

97

81.6

Methanol

0.762

128

64.6

  1. Line 94, Page 2. “they were” should be “it was”.

Answer:

We thank the reviewer for this observation. The mistake was amended, and the whole manuscript was revised

  1. Line 175, Page 5. What will happen if the substrate is treated with plasma to enhance the wetting with the droplet of acetonitrile?

Answer:

It is known that plasma treatment improves the wettability of surfaces and can change the hydrophobic behavior to a hydrophilic one on several surfaces. A similar phenomenon could be expected for acetonitrile solutions. If wettability is improved, a homogeneous film could be obtained for the acetonitrile solution. Unfortunately, we cannot conduct the experiment at this moment, but we will consider it in the future. We thank the reviewer for this important observation.

  1. Line 210. Do you use the solvent of chloroform?

Answer:

We apologize for this mistake; the reviewer is right chloroform was not used. The conclusions section mentions chloroform, but the correct solvent is chlorobenzene. The error

was amended. We thank the reviewer again for this comment that improves our manuscript.

Reviewer 4 Report

Achieving high-quality films is essential for the performance of optoelectronic devices. In the draft, the authors studied the solvent effect on the small molecule thin films deposited by the doctor-blade method. The morphology of thin films with solvents deposited at varied temperatures was demonstrated and analyzed. The draft can be accepted by solving the following problem:

1. In table one, it is better the  authors also listed out the boiling temperature, polarity( dielectric constant), and viscosity of each solvent

2. In Figure1 and 2, what is the size and magnification of the inserted figure under UV?

3.DMF in the draft was written as DFM in some texts and the title of figure 2

Author Response

Reviewer 4

Comments and Suggestions for Authors

Achieving high-quality films is essential for the performance of optoelectronic devices. In the draft, the authors studied the solvent effect on the small molecule thin films deposited by the doctor-blade method. The morphology of thin films with solvents deposited at varied temperatures was demonstrated and analyzed. The draft can be accepted by solving the following problem:

  1. In table one, it is better the authors also listed out the boiling temperature, polarity( dielectric constant), and viscosity of each solvent

Answer:

According to the reviewer's comment, on page 4, Table 1 was modified by adding the relative polarity, vapor pressure, and boiling point for the used solvents. The added values are:

 Physicochemical parameters of the used solvents.

Solvent

Relative polarity

Vapor pressure

(hPa)

Boiling point

(°C)

Heptane

0.012

48

98

Chlorobenzene

0.188

12

132

DMF

0.386

3.5

153

Acetonitrile

0.46

97

81.6

Methanol

0.762

128

64.6

  1. In Figure1 and 2, what is the size and magnification of the inserted figure under UV?

Answer:

The pictures of films inserted in the AFM images are of the 5x5 cm substrates. A sizing bar was added to the pictures to show their size.

3.DMF in the draft was written as DFM in some texts and the title of figure 2

Answer:

We apologize for this mistake; the whole manuscript was revised, and the error was amended. We thank the reviewer.

Round 2

Reviewer 3 Report

The manuscript can be accepted now.